



# A comparative analysis for deep learning model (hyDL-CO v1.0) and Kalman Filter to predict CO in China

Weichao Han[1]†, Tai-Long He[2]†, Zhaojun Tang[1], Min Wang[1], Dylan Jones[2], Zhe Jiang[1]*

[1]School of Earth and Space Sciences, University of Science and Technology of China, Hefei, Anhui, 230026, China.
[2]Department of Physics, University of Toronto, Toronto, ON, M5S 1A7, Canada.
†These authors contributed equally to this work.

*Correspondence to: Zhe Jiang (zhejiang@ustc.edu.cn)

## Abstract

The applications of novel deep learning techniques in atmospheric science are rising quickly. Here we build a hybrid deep learning (DL) model (hyDL-CO), based on convolutional neural networks (CNN) and long short-term memory (LSTM) neural networks to provide a comparative analysis between DL and Kalman Filter (KF) to predict carbon monoxide (CO) concentrations in China in 2015-2020. We find the performance of DL model is better than KF in the training period (2015-2018): the mean bias and correlation coefficients are 9.6 ppb and 0.98 over E. China, and -12.5 ppb and 0.96 over grids with independent observations. By contrast, the assimilated CO concentrations by KF exhibit comparable correlation coefficients but larger negative biases. Furthermore, DL model demonstrates good temporal extensibility: the mean bias and correlation coefficients are 95.7 ppb and 0.93 over E. China, and 81.0 ppb and 0.91 over grids with independent observations in 2019-2020, while CO observations are not fed into the DL model as an input variable. Despite these advantages, our analysis indicates a noticeable underestimation of CO concentrations at extreme pollution events in the DL model. This work demonstrates the advantages and disadvantages of DL models to predict atmospheric compositions in respective to traditional data assimilation, which is helpful for better applications of this novel technique in future studies.



## 1. Introduction

Accurate simulation and prediction of air pollutants are critical for making effective policies to improve air quality. Chemical transport models (CTMs), as powerful tools, have been widely used to simulate atmospheric compositions (Li et al., 2019; Chen, X. et al., 2021; Lu et al., 2021). Despite the advances of CTMs, there are still noticeable discrepancies in the simulations due to uncertainties in the emission, physical and chemical processes (Quennehen et al., 2016; Kong et al., 2020). Atmospheric observations are thus used to evaluate the capacity of CTMs to capture the observed variabilities in atmospheric composition. For example, Liu et al. (2018) found the modeled spatial variability of nitrogen dioxides ($NO_2$) matches well with surface observations but with a large bias in their concentrations. Zhang et al. (2021) exhibited a difference between modeled and observed surface fine particulate matter ($PM_{2.5}$) and ozone ($O_3$): the modeled $PM_{2.5}$ and $O_3$ concentrations are higher than observations by about 40% and 15% in China in 2013-2017, respectively.

Based on CTMs, data assimilation techniques integrate simulations and observations and thus can improve the modeled atmospheric compositions. For instance, Feng et al. (2018) found the assimilation of surface $PM_{2.5}$ observations can effectively reduce the uncertainties in $PM_{2.5}$ forecasts. Peng et al. (2018) assimilated surface observations, including $PM_{2.5}$, $NO_2$, $O_3$, CO, and obtained near-perfect forecasts on the first day, but the effects of the data assimilation decayed quickly with longer forecasts. The propagation of observational information in data assimilation depends on the modeled physical and chemical processes, i.e., the adjustment over grids lacking observations relies on regional transport of observational information from other grids. The assimilated results are thus, still affected by potential model errors (e.g., the uncertainty in transport), which can lead to rapid decline of assimilation effects, if observations become unavailable.

Accompanied with recent advance of machine learning (ML) techniques, novel data-



driven architectures and approaches have been extensively applied in the field of atmospheric
science (Li et al., 2020; Zhang et al., 2020; Shi et al., 2021; Xing et al., 2021). Based on
artificial neural networks, particularly, CNNs, DL uses multiple layers of computational
kernels to extract and capture non-linear relationships between input and output variables. The
predictions, provided by DL, are driven by observational or reanalysis data sets, which provides
a new way to predicting atmospheric compositions without the influences from model errors.
The non-linear relationships learned in the training data set can be extended spatially and
temporally, for example, Kleinert et al. (2021) found the DL model can forecast surface $O_3$
within a 4-day range. The application of LSTM networks further improves the ability of DL
models in capturing temporal dynamics, for example, Chen, Y. et al. (2021) found the LSTM-
based approach can provide a good prediction for surface $PM_{2.5}$ on the next day; He et al. (2021)
exhibited the capability of DL model to predict surface $O_3$ in the North America.

Despite the advantages of the DL approaches, the lack of parameterization of physical

and chemical processes implies the predicted atmospheric compositions may deviate from the
realistic atmospheric state, in contrast to conventional data assimilation approaches that are
constrained by modeled processes. Tropospheric CO is one of the most important pollutants
with significant sources from fossil fuel combustion. Because of the lifetime (about 1-2
months), tropospheric CO is an ideal tracer for atmospheric transport and has been sufficiently
investigated with data assimilations (Feng et al., 2020; Peng et al., 2018; Tang et al., 2021). In
this study, we present an application of a hybrid DL model (hyDL-CO) on prediction of surface
CO in China from 2015 to 2020, which utilizes both CNNs and LSTMs. We perform a
comparative analysis between the DL model and a KF system in this work, to investigate the
performances of the two approaches in predicting CO. This comparison is helpful for
understanding the advantages and disadvantages of the DL approach in respective to traditional
data assimilation, which is critical for better applications of this novel technique in atmospheric





environmental studies in the future.
This paper is organized as follows: in Section 2, we describe the CO observations, the
KF approach and the hyDL-CO model used in this work. In Section 3, we assess the predicted
CO by the DL model, the changes in CO emissions in China, as well as the comparison between
the DL model and KF, and the evaluation with independent observations. Our conclusions
follow in Section 4.

**2. Data and Methodology**
**2.1 MEE surface CO measurements**
We use the China Ministry of Ecology and Environment (MEE) monitoring network
surface in-situ CO concentration data (https://quotsoft.net/air/) for the period of 2015–2020.
These real-time monitoring stations have the ability to report hourly concentrations of criteria
pollutants from about 1700 sites in 2020. Concentrations were reported by the MEE in units of
ug/m$^3$ under standard temperature (273 K) until 31 August 2018. This reference state was
changed on 1 September 2018 to 298 K. We converted CO concentrations to ppb and rescaled
post-August 2018 concentrations to standard temperature (273 K) to keep the consistency in
the trend analysis. The reported data with CO concentrations larger than 6000 ppb are removed
in our analysis. The station-based observations are averaged and regrided to the 0.5°x0.625°
grid of the MERRA-2 reanalysis using the nearest neighborhood interpolation algorithm, with
totally about 500 grids having observations. 10% grid-based observations (about 50 grids) are
randomly selected as independent observations, which are only used in the evaluation of the
predicted CO from the DL model and the KF system. The training of the DL model and the
assimilation using the KF are performed using the remaining 90% observations.

**2.2 KF approach**
We employ the sequential KF based on the GEOS-Chem CTM to assimilate surface CO
observations. This approach has been used in previous studies to optimize tropospheric CO



concentrations (Jiang et al., 2017; Tang et al., 2021). The GEOS-Chem model
(http://www.geos-chem.org, version 12-8-1) is driven by assimilated meteorological data of
MERRA-2. Our analysis is conducted at a horizontal resolution of nested 0.5°x0.625° and
employs the CO-only simulation in GEOS-Chem, which uses archived monthly OH fields from
the full chemistry simulation (Fisher et al., 2017). The CO boundary conditions are updated
every 3-hour from a global simulation with $4° \times 5°$ resolution. Emissions in GEOS-Chem are
computed by the Harvard-NASA Emission Component (HEMCO). Global default
anthropogenic emissions are from the Community Emissions Data System (CEDS) (Hoesly et
al., 2018) and replaced by MEIC (Multiresolution Emission Inventory for China) in China and
MIX (full name) in other regions of Asia (Li et al., 2017). The total anthropogenic CO
emissions in MEIC inventory are further scaled with linear projection. We refer the reader to
Chen, X. et al. (2021) for the details of model configurations.
In the assimilation algorithm, the forward model ($M$) predicts CO concentration ($x_{at}$) at
time $t$:
$$x_{at} = M_t x_{t-1} \qquad \text{(Eq. 1)}$$
The optimized CO concentrations can be expressed as:
$$x_t = x_{at} + G_t(y_t - K_t x_{at}) \qquad \text{(Eq. 2)}$$
where $y_t$ is observation, $K_t$ represents operation operator which projects CO concentrations
from the model space to observation space. $G_t$ is the KF Gain matrix, which can be described
as:
$$G_t = S_{at} K_t^T (K_t S_{at} K_t^T + S_\epsilon)^{-1} \qquad \text{(Eq. 3)}$$
where $S_{at}$ and $S_\epsilon$ are model and observation covariance, respectively. Because the DL
model is designed to reproduce observations without considering error covariance, here we
assume fixed model error (50%) and small observation error (1%) to provide a fair comparison.
The covariance matrix is diagonal without the consideration of off-diagonals.

**2.3 hyDL-CO v1.0 model**





We combine CNN and LSTM to obtain a hybrid model for the prediction of surface CO
in China, following He et al. (2021). As shown in Fig. 1, the hyDL-CO model is an autoencoder
with the latent space represented by a LSTM cell. The first three blocks of neural layers behave
as an encoder, which has six convolutional layers and two max pooling layers, to extract the
features hidden in the input data. A dropout layer is added after each pooling layer to prevent
data overfitting. The output from the encoder is highly compressed information that is not
manipulated during the training process, which is also called the latent vector. We embed the
LSTM model into the DL architecture after the encoder to capture short-term changes and long-
term trends in the latent vectors. The output from the LSTM is then forwarded to a decoder
with three blocks of layers. Each block in the decoder has one transposed convolutional layer
followed by two convolutional layers. The outputs from each convolutional layer in the model
are passed through the Rectified Linear Unit (ReLU) activation function to increase non-
linearity. Residual learning connections that forward the high-resolution features extracted by
the encoder to the decoder are also added, which are shown to improve the performance of the
DL model (Ronneberger et al., 2015; He et al., 2015). These connections contain trainable
weights that represents more direct relationship between input and output variables.
The optimization of the model is supervised by the "ground truth", which is the daily
mean surface CO concentrations measured by the MEE network. The weights in the CNNs and
transposed CNNs are optimized using the back-propagation algorithm (Rumelhart et al., 1986;
LeCun et al., 1989), which employs the partial derivatives of cost function with respect to the
truth. The loss function to be optimized is the mean square error (MSE) between the "predicted"
and "true" values. We use the Adam optimizer, which is a computationally efficient algorithm
for gradient-based optimization of stochastic objective functions. For a faster convergence
speed and the stability of the model performance, we rescale all the features to a nearly same
scale. The processing method is multiplying the original variable by a constant $10^n$ and adapting
$n$ for each variable according to the specified scale. This processing prevents the DL model to
be overfit by the features in input variables that have significantly larger scales than others.
The hybrid model was built and implemented using Keras and Tensorflow, which are Python



packages that are extensively used in DL studies. Table 1 shows some of the configuration
hyperparameters of the training of our model.

The input variables include six meteorological variables: sea level pressure (SLP),

surface incoming shortwave flux (SWGDN), 2-meter air temperature (T2M), 10-meter
eastward wind (U10M), 10-meter northward wind (V10M) and total precipitation (TP); and
total anthropogenic CO and volatile organic compounds (VOC) emissions. The meteorology
and emission data are extracted from the GEOS-Chem model with 0.5°x0.625° horizontal
resolution. Our focus area is 0-72°N, 0-180°E, and the output resolution is same as the
0.5°x0.625° resolution of MERRA-2. The DL model grid thus has 288 grid boxes along the
longitudinal direction and 144 for the latitude. Considering the long lifetime of CO, the
concentration of surface CO is not only related to the emission and meteorological conditions
at the current moment, but also at the previous moment. We trained the DL model using the
information related to the "history" of CO, by adding the same set of input variables for the
current day and previous four days as predictors. The information from the 5-day history has
40 predictors in total for the prediction of daily mean surface CO in each day. We use 2015-
2018 as the training data set and 2019-2020 as the test set. The dimension of each input vector
for the DL model is then (144,288,40), and the dimension of the output from the DL model is

(144,288,1).


**3. Results and Discussions**
**3.1 CO concentrations predicted by DL model**

As shown in Fig. 2A, the annual averaged MEE CO observations are broadly higher than

400 ppb in E. China in 2015-2018 and can reach 1000 ppb over highly polluted North China
Plain (NCP). The predicted CO concentrations by the DL model (Fig. 2B) match well with
observations in 2015-2018. We find small differences between predictions and observations in
Fig. 2C. The Pearson correlation coefficients are larger than 0.7 over E. China and are as high
as 0.9 over highly polluted NCP (Fig. 2D). Fig. 3A-E exhibit daily variabilities of CO



concentrations over E. China, as well as NCP, Yangtze River Delta (YRD), Pearl River Delta
(PRD) and Sichuan Basin (SCB) domains. There is large seasonality in the observed CO
concentrations: the wintertime CO concentrations can reach 1400 ppb over E. China, and 2500
ppb over highly polluted NCP; the summertime CO concentrations are about 500 ppb over E.
China and 800 ppb over NCP. The predicted CO concentrations by the DL model demonstrate
high consistency with observations. As shown in Table 2, the correlation coefficients between
DL model and MEE CO observations are 0.98, 0.97, 0.93, 0.89 and 0.90; the biases are 9.6,
18.2, -2.6, 12.7 and 17.6 ppb for E. China, NCP, YRD, PRD and SCB, respectively.

The high consistency between observations and DL model in the training period (2015-

2018) is expected. Here we further evaluate the capability of DL model to predict CO
concentrations without the inputs of CO observations (i.e., in the test period). Fig. 2E shows
the MEE CO observations in 2019-2020. As shown in Fig. 2F, the DL model overestimated
surface CO concentrations in 2019-2020, particularly, over highly polluted NCP. The Pearson
correlation coefficients in 2019-2020 (Fig. 2H) are slightly lower than those in the training
period (Fig. 2D). As shown in Fig. 3F-J, the predicted CO concentrations exhibit larger
deviations from observations in 2019-2020. The correlation coefficients (See Table 2) between
observed and predicted CO in the test period are 0.93, 0.92, 0.81, 0.80 and 0.83; the biases are
95.7, 224.2, 22.0, 60.8 and 52.8 ppb for E. China, NCP, YRD, PRD and SCB, respectively.
Consequently, the lack of inputs of CO observations in the test period led to a decline of
prediction capability, but it is still high enough to provide useful information to predict CO
variabilities.
**3.2 Changes of CO emissions inferred by DL model**

As shown in Fig. 3F, the predicted CO concentrations by DL model show large difference

with observations in 2019-2020, by contrast, there is good agreement in 2015-2018 (Fig. 3A).
The observed CO concentrations are about 650 ppb in the summer of 2015 and decreased



gradually to about 600 ppb by the summer of 2018. However, the observed CO concentrations
dropped to about 550 ppb in the summer of 2019 and 2020. The rapid decrease of surface CO
concentrations is dominated by highly polluted NCP (Fig. 3G), whereas the differences
between predicted and observed CO concentrations are limited over other domains. It seems
that the rapid decrease of surface CO concentrations over NCP 2019 is associated with an
unexpected drop in CO emissions, which is not considered in the linear projection of emission
inventory, and led to overestimated CO concentrations in the DL model.
The unprecedented lockdowns across the world to contain the 2019 novel coronavirus
(COVID-19) spread have led to a slowdown of economic activities, with pronounced declines
in anthropogenic emissions. Shi and Brasseur (2020) found surface CO concentrations over N.
China were 1.2-1.5 and 0.7-1.0 mg/m$^3$ before and during the pandemic spread. Gaubert et al.
(2021) suggested about 15% reduction in CO emissions over N. China due to the COVID-19
controls. As shown in Fig. 3F, the MEE CO observations match well with predicted CO by DL
model in early 2019, however, are much lower than the predicted CO in early 2020. By contrast,
the difference between observed and predicted CO concentrations are comparable in the
summer of 2019 and 2020. The large discrepancy between observations and predictions in early
2020 thus, reflects the decline of CO emissions caused by COVID-19 controls, which is not
considered in the linear projection of emission inventory.
**3.3 Comparison between DL model and KF assimilation**
Fig. 2I-P show the MEE CO observations and assimilated CO concentrations by KF in
2015-2018 and 2019-2020, respectively. While the spatial distributions of assimilated CO
match well with observations, the CO concentrations in the assimilations are noticeably lower.
As shown in Fig. 3A-E and Table 2, the differences between assimilated and observed CO are
-114.9, -139.6, -58.0, -108.8 and -29.3 ppb for E. China, NCP, YRD, PRD and SCB,
respectively, which are larger than the differences in the DL model. Furthermore, the modeled



CO concentrations in the control runs (CR, without assimilation of CO observations) are much
lower: the differences are -409.6, -512.3, -246.0, -400.5 and -172.4 ppb for E. China, NCP,
YRD, PRD and SCB, respectively. The dramatic underestimations of CO concentrations in
model simulations have been reported in recent studies (Feng et al., 2020; Peng et al., 2018),
which could be associated with significant model representation error because most MEE
stations are urban sites (Tang et al., 2021). It reveals the important discrepancy between DL
and data assimilations: the analyzed concentrations in KF are based on the a priori and observed
concentrations by considering the model and observation errors, which is not designed to
reproduce the observations. In addition, the correlation coefficients are 0.99, 0.99, 0.98, 0.94
and 0.96 for E. China, NCP, YRD, PRD and SCB in 2015-2018 in the KF, respectively, which
are comparable with the DL model.

As shown in Fig. 3F-J and Table 2, the difference between assimilated and observed CO

concentrations in 2019-2020 are -85.5, -66.3, -52.9, -89.3 and -18.7 ppb for E. China, NCP,
YRD, PRD and SCB, respectively, which are comparable the differences in DL model except
highly polluted NCP, even the MEE CO observations are not inputted in DL model in the test
period. The correlation coefficients are 0.99, 0.99, 0.97, 0.96 and 0.96 for E. China, NCP, YRD,
PRD and SCB in 2019-2020 in the KF, respectively, which are higher than the DL model. In
addition, Fig. 4A-B show the relationships between modeled CO and MEE CO observations.
Both DL and KF show dramatic improvements in respective to the CR simulations in Fig. 4A-
B, while the performance of the DL model is better than KF in the training period (Fig. 4A). In
addition, the comparable performances between DL and KF in 2019-2020 (Fig. 4B)
demonstrate the good temporal extensibility of DL model, i.e., skills learned in the training
period can be extended to the following years with a limited decline in the prediction effects.
**3.4 Evaluation with independent MEE CO observations**

Fig. 5A-B show the spatial distributions of predicted CO concentrations by DL model



and MEE CO observations; Fig. 6A-B further exhibit the locations of randomly selected
independent MEE stations (about 10% of total stations). These independent stations are not
used in both DL model and KF in 2015-2020. We find good agreements between predicted CO
concentrations by DL model and MEE CO observations. The DL model suggests the highest
CO concentrations in the Shanxi province, by more than 1200 ppb, and background CO
concentrations by about 400 ppb over remote areas. By contrast, the CO concentrations in the
KF (Fig. 5C-D; Fig. 6C-D) are lower, and the highest CO concentrations are found in NCP
rather than Shanxi province. As shown in Fig. 7A-E, the DL model demonstrates smaller bias
in respective to independent MEE CO observations and higher correlation coefficients than KF
in 2015-2018, suggesting better capability to predict CO concentrations. In 2019-2020 (Fig.
7F-J), the DL model exhibits a smaller bias over E. China, but larger bias than KF over highly
polluted NCP. The Pearson correlation coefficients are smaller in DL in 2019-2020 (See Table

2).

As shown in Fig. 4C-D, the assimilated CO concentrations by KF are closer to the control
simulations with larger deviations from the MEE CO observations than those in Fig. 4A-B. It
demonstrates the decline of assimilation effects when observations are unavailable. On the
other hand, the slopes in the linear fits are 0.89 and 0.92 in DL and KF in 2015-2018 (Fig. 4C),
respectively, and become 0.80 and 1.02 in 2019-2020 (Fig. 4D). The deviations in the slopes
reflect an underestimation of CO concentrations in the DL model at grids with extremely high
CO concentrations. DL model predicts CO concentrations based on the skills learned in the
training process. However, the training is dominated by the majority of CO observations with
low and medium CO concentrations, while the extreme high CO concentrations (i.e., extreme
pollution events) cannot be learned sufficiently. By contrast, KF is driven by observations
directly, and thus, both high and low CO concentrations can be simulated. In addition, because
most MEE stations are urban sites, the good agreement between DL model and MEE CO



observations may not be able to ensure the accuracy of predicted CO concentrations over
remote rural areas. Integration of modeled CO concentrations in the DL model in future studies
may improve predicted CO concentrations over remote areas without local observations.
**4. Conclusion**

294   A hybrid DL model (hyDL-CO), based on CNN and LSTM, was built in this work to

provide a comparative analysis between DL and KF to predict CO concentrations in China in
2015-2020. We find the performance of the DL model is better than KF in the training period
(2015-2018): the bias and correlation coefficients are 9.6 ppb and 0.98 over E. China, and -
12.5 ppb and 0.96 over grids with independent observations. By contrast, the assimilated CO
concentrations by KF demonstrate comparable correlation coefficients but larger negative
biases: the bias and correlation coefficients are -114.9 ppb and 0.99 over E. China, and -252.5
ppb and 0.95 over grids with independent observations. The larger biases in the KF are caused
by the discrepancy in the algorithm, i.e., the objective of data assimilation is to improve the
simulated atmospheric compositions by considering the model and observation errors, which
is not designed to reproduce the observations. Both DL and KF show better predictions than
the control runs: the bias and correlation coefficients are -409.6 ppb and 0.94 over E. China,
and -443.3 ppb and 0.91 over grids with independent observations.

307   Furthermore, we find good temporal extensibility of the DL model in the test period

(2019-2020): the bias and correlation coefficients are 95.7 ppb and 0.93 over E. China, and
81.0 ppb and 0.91 over grids with independent observations. The correlation coefficients (0.91-
0.93) mean enough capability to provide useful information to predict CO variabilities without
inputs of CO observations. In addition, we find an unexpected drop of CO emissions over
highly polluted NCP in 2019. Our analysis further exhibits a significant decline of CO
emissions in early 2020 due to the COVID-19 controls. Despite these advantages, we find
noticeable underestimation of CO concentrations at grids with extreme high CO concentrations



in the DL model, because the training is dominated by the majority of CO observations with
low and medium CO concentrations, and thus, the extreme pollution events cannot be learned
sufficiently. This work demonstrates the advantages and disadvantages of DL models to predict
atmospheric compositions in respective to traditional data assimilation. We advise more efforts
to explore new applications of DL models in atmospheric environmental studies.

**Code and data availability:** The MEE CO data can be downloaded from
https://quotsoft.net/air/. The GEOS-Chem model (version 12.8.1) can be downloaded from
http://wiki.seas.harvard.edu/geos-chem/index.php/GEOS-Chem_12#12.8.1. The code of the
hyDL-CO model, sample data for the hyDL-CO model run and GEOS-Chem model output can
be downloaded from https://doi.org/10.5281/zenodo.5913013.

**Author Contributions**: Z.J. designed the research. W.H. and T.-L.H. developed the model
code and performed the research. Z.J. wrote the manuscript. All authors contributed to
discussions and editing the manuscript.

**Competing interests**: The authors declare that they have no conflict of interest.

**Acknowledgments:** We thank the China Ministry of Ecology and Environment (MEE) for
providing the surface CO measurements. The numerical calculations in this paper have been
done on the supercomputing system in the Supercomputing Center of University of Science
and Technology of China. This work was supported by the Hundred Talents Program of
Chinese Academy of Science and National Natural Science Foundation of China (41721002).
**Table and Figures**
**Table 1.** Hyperparameters used in the hybrid DL model.





**Table 2.** Deep learning (DL), Kalman Filter (KF) and control run (CR) in respective to MEE
CO observations in 2015-2018 and 2019-2020. The locations of independent MEE stations are
shown in Fig. 6.

**Figure 1.** Hybrid DL model used in this paper.

**Figure 2.** (A) MEE CO observations in 2015-2018; (B) Predicted CO concentrations by DL
model in 2015-2018; (C-D) differences and Pearson correlation coefficients between predicted
and observed CO in 2015-2018. (E-H) MEE CO observations, predicted CO concentrations by
DL model and their differences, and Pearson correlation coefficients in 2019-2020. (I-P) Same
as panels A-H, but for KF. The unit is ppb.

**Figure 3.** Daily variabilities of CO concentrations from MEE, DL and KF in 2015-2018 and

354  2019-2020.


**Figure 4.** (A-B) Relationships between CO concentrations provided by DL, KF, control run
(CR) and MEE CO observations in 2015-2018 and 2019-2020. The dots represent daily average
of CO concentrations over E. China. The unit is ppb. (C-D) Same as panels A-B, but with
randomly selected independent MEE stations. The locations of independent MEE stations are
shown in Fig. 6.

**Figure 5.** (A-B) Predicted by DL (contour) and MEE (dotted) surface CO concentrations in
2015-2018 and 2019-2020; (C-D) Same as panels A-B, but for KF.

**Figure 6.** (A-B) Predicted by DL (contour) and independent MEE (dotted) surface CO
concentrations in 2015-2018 and 2019-2020; (C-D) Same as panels A-B, but for KF. The
randomly selected independent MEE stations (about 10% of total stations) are not used in both
DL and KF in 2015-2020.

**Figure 7.** Daily variabilities of CO concentrations from independent MEE stations, DL and KF
in 2015-2018 and 2019-2020. The locations of independent MEE stations are shown in Fig. 6.



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





| Optimizers | Learning rate | EarlyStopping patience | Batch size | Epochs | Validation split | shuffle |
|---|---|---|---|---|---|---|
| Adam | 0.001 | 20 | 64 | 500 | 0.125 | True |

**Table. 1.** Hyperparameters used in the hybrid DL model.

| | | | 90% MEE stations | | | | | Independent MEE stations | | | | |
|---|---|---|---|---|---|---|---|---|---|---|---|---|
| | | | E. China | NCP | YRD | PRD | SCB | E. China | NCP | YRD | PRD | SCB |
| **2015-2018 ( Training period )** | **Bias (ppb)** | DL | 9.6 | 18.2 | -2.6 | 12.7 | 17.6 | -12.5 | 92.3 | -29.9 | -19.8 | -280.4 |
| | | KF | -114.9 | -139.6 | -58.0 | -108.8 | -29.3 | -252.5 | -122.2 | -165.6 | -208.0 | -141.7 |
| | | CR | -409.6 | -512.3 | -246.0 | -400.5 | -172.4 | -443.3 | -403.9 | -319.2 | -402.3 | -371.6 |
| | **R** | DL | 0.98 | 0.97 | 0.93 | 0.89 | 0.90 | 0.96 | 0.94 | 0.86 | 0.73 | 0.78 |
| | | KF | 0.99 | 0.99 | 0.98 | 0.94 | 0.96 | 0.95 | 0.91 | 0.84 | 0.66 | 0.76 |
| | | CR | 0.94 | 0.87 | 0.83 | 0.68 | 0.78 | 0.91 | 0.79 | 0.74 | 0.58 | 0.64 |
| | **Slope** | DL | 0.95 | 0.91 | 0.80 | 0.73 | 0.78 | 0.89 | 0.89 | 0.70 | 0.46 | 0.49 |
| | | KF | 1.02 | 0.98 | 1.04 | 0.99 | 1.07 | 0.92 | 1.06 | 0.93 | 0.68 | 0.84 |
| | | CR | 0.71 | 0.63 | 0.92 | 0.58 | 1.26 | 0.72 | 0.78 | 0.72 | 0.44 | 0.83 |
| | | | E. China | NCP | YRD | PRD | SCB | E. China | NCP | YRD | PRD | SCB |
| **2019-2020 ( Test period )** | **Bias (ppb)** | DL | 95.7 | 224.2 | 22.0 | 60.8 | 52.8 | 81.0 | 237.1 | 1.9 | 60.4 | -57.6 |
| | | KF | -85.5 | -66.3 | -52.9 | -89.3 | -18.7 | -167.1 | -46.9 | -144.0 | -127.2 | 75.6 |
| | | CR | -279.7 | -202.1 | -194.0 | -328.7 | -69.3 | -297.8 | -168.1 | -262.7 | -299.8 | -49.9 |
| | **R** | DL | 0.93 | 0.92 | 0.81 | 0.80 | 0.83 | 0.91 | 0.84 | 0.77 | 0.74 | 0.70 |
| | | KF | 0.99 | 0.99 | 0.97 | 0.96 | 0.96 | 0.96 | 0.89 | 0.85 | 0.79 | 0.75 |
| | | CR | 0.94 | 0.89 | 0.77 | 0.76 | 0.79 | 0.91 | 0.78 | 0.76 | 0.74 | 0.67 |
| | **Slope** | DL | 0.90 | 0.95 | 0.70 | 0.65 | 0.80 | 0.79 | 0.86 | 0.57 | 0.42 | 0.54 |
| | | KF | 1.05 | 1.02 | 1.05 | 1.02 | 1.14 | 1.02 | 1.21 | 0.96 | 0.82 | 1.13 |
| | | CR | 0.96 | 0.97 | 1.04 | 0.71 | 1.81 | 0.93 | 1.14 | 0.84 | 0.60 | 1.45 |

**Table. 2.** Deep learning (DL), Kalman Filter (KF) and control run (CR) in respective to MEE CO observations in 2015-2018 and 2019-2020. The locations of independent MEE stations are shown in Fig. 6.



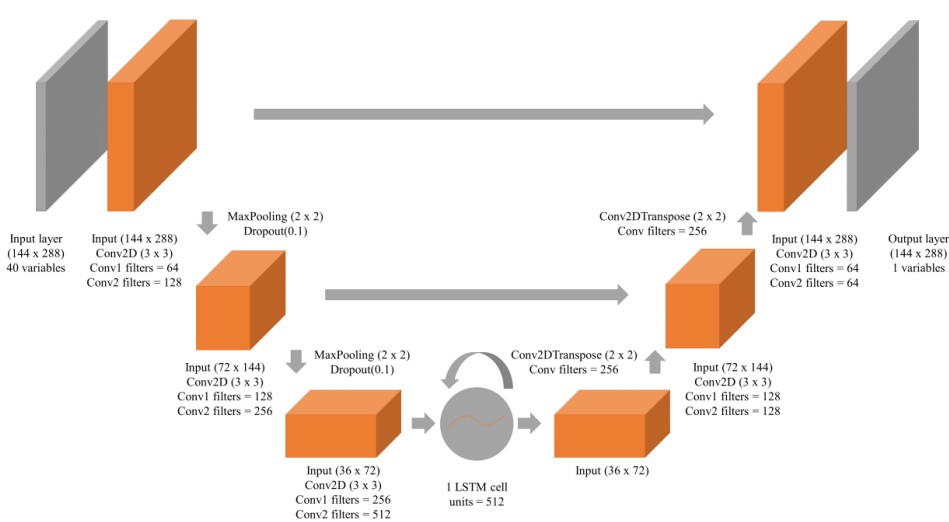

**Fig. 1.** Hybrid DL model used in this paper.



**Fig. 2.** (A) MEE CO observations in 2015-2018; (B) Predicted CO concentrations by DL model in 2015-2018; (C-D) differences and Pearson correlation coefficients between predicted and observed CO in 2015-2018. (E-H) MEE CO observations, predicted CO concentrations by DL model and their differences, and Pearson correlation coefficients in 2019-2020. (I-P) Same as panels A-H, but for KF. The unit is ppb.



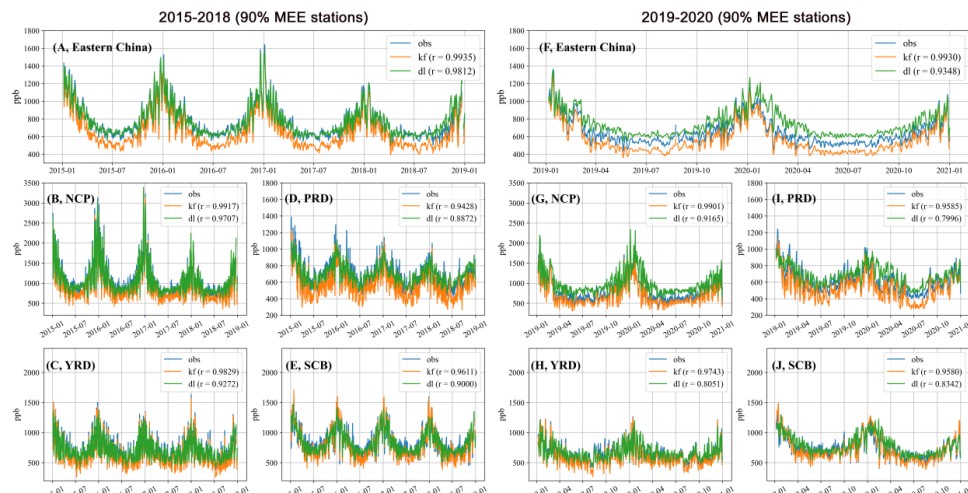

**Fig. 3.** Daily variabilities of CO concentrations from MEE, DL and KF in 2015-2018 and 2019-2020.





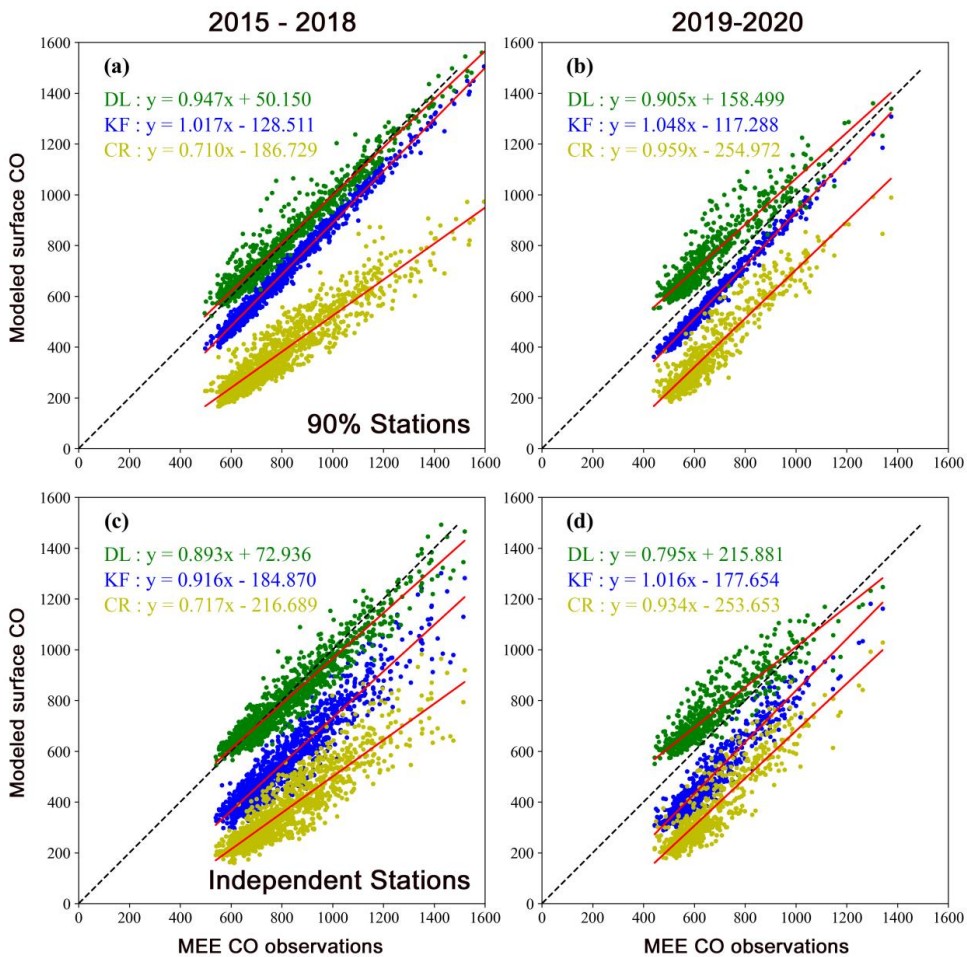

**Fig. 4.** (A-B) Relationships between CO concentrations provided by DL, KF, control run (CR) and MEE CO observations in 2015-2018 and 2019-2020. The dots represent daily average of CO concentrations over E. China. The unit is ppb. (C-D) Same as panels A-B, but with randomly selected independent MEE stations. The locations of independent MEE stations are shown in Fig. 6.



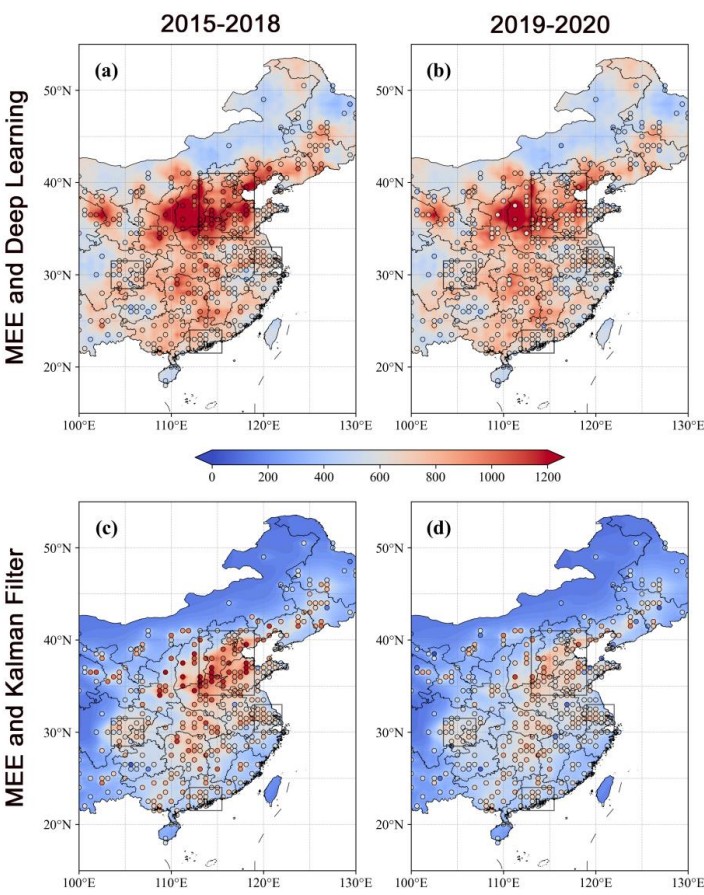

**Fig. 5.** (A-B) Predicted by DL (contour) and MEE (dotted) surface CO concentrations in 2015-2018 and 2019-2020; (C-D) Same as panels A-B, but for KF.



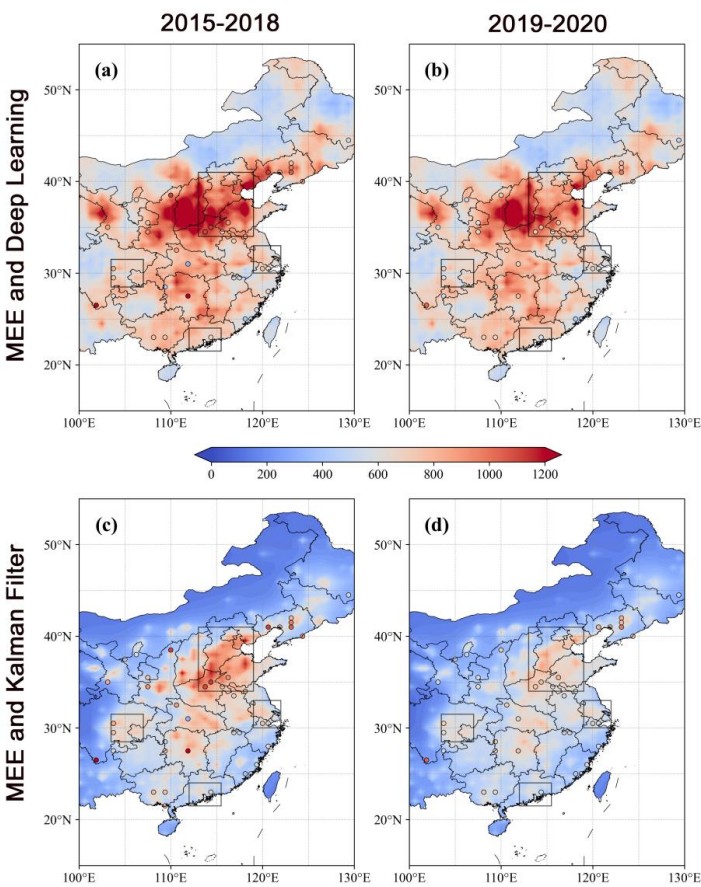

**Fig. 6.** (A-B) Predicted by DL (contour) and independent MEE (dotted) surface CO concentrations in 2015-2018 and 2019-2020; (C-D) Same as panels A-B, but for KF. The randomly selected independent MEE stations (about 10% of total stations) are not used in both DL and KF in 2015-2020.





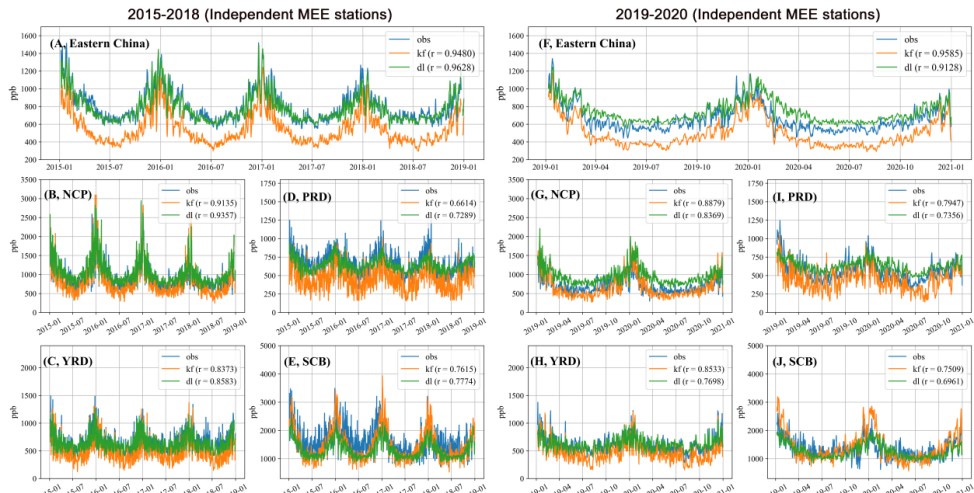

**Fig. 7.** Daily variabilities of CO concentrations from independent MEE stations, DL and KF in 2015-2018 and 2019-2020. The locations of independent MEE stations are shown in Fig. 6.