# Peer review of "A comparative analysis for a deep learning model (hyDL-CO v1.0) and Kalman Filter to predict CO concentrations in China"

_Geoscientific Model Development, 2021_

## Author Response (AR1)

We thank the reviewers for their thoughtful and detailed comments. We have revised this manuscript based on the comments. Below we respond to the individual comments.

Reviewer #1:

The paper compared the performances of a deep learning model and a chemical transport model with data assimilation in predicting surface carbon monoxide in China. It is of high scientific significance, as the results have implications for understanding the roles of machine learning and numerical methods in the prediction of air pollution. The manuscript is well-written. I recommend the publication of the paper with minor revision. I have the following comments that need the authors to address.

**Answer**: Thank you for the comments! The manuscript has been revised based on the comments.

**Question**: (1) How does the deep learning model, hyDL-CO, developed in this study perform when compared with other deep learning models?

Answer: A new paragraph (at the beginning of Section 2.3) and more discussions were provided in the revision: "Our hyDL-CO model is a modified version of the U-net model used in He et al. (2022), where the model shows a superior capability in predicting surface summertime O3 in North America. The U-net architecture is a variant of autoencoder and was originally proposed for biomedical segmentation applications. In the first U-net paper, Ronneberger et al. (2015) conducted three experiments and showed that the U-net model outperforms other DL models. Since the proposal of U-net, it has become one of the most popular choices in the DL community and is compared with other ML models in many studies. For example, Korznikov et al. (2021) used several ML models for tree recognition using satellite images and the U-net model shows the highest accuracy. Ravuri et al. (2021) used U-net as a baseline model and compared against their Generative Adversarial Network (GAN) in precipitation nowcasting. Andersson et al. (2021) showed that their IceNet, which is an ensemble of similar U-Net networks, has outstanding performance in seasonal forecasts of Arctic Sea ice".

**Question**: (2) I recommend the authors use more indices to evaluate the performances of the models. The Figures and Tables show that the Kalman Filter performs better than the deep learning model in the test period of 2019-2020. I think this is important that need to be mentioned in the abstract.

**Answer**: Thank you for this suggestion! Root Mean Square Error (RMSE) was added in Table 2. The following sentence was added in the abstract: "we find a weaker prediction capability of DL model than KF in the test period".

**Question**: (3) How do models perform in simulating the spatial variability of surface carbon monoxide?

**Answer**: As shown in the revised manuscript: "Although we find broadly good agreements in the spatial distributions between predicted CO concentrations by DL and KF and MEE CO observations, there is still a noticeable discrepancy. The DL model suggests the highest CO concentrations in the Shanxi province, by more than 1200 ppb, and background CO concentrations by about 400 ppb over remote areas. By contrast,

the CO concentrations in the KF (Fig. 5C-D; Fig. 6C-D) are lower, and the highest CO concentrations are found in NCP rather than Shanxi province". "because most MEE stations are urban sites, the good agreement between DL model and MEE CO observations may not be able to ensure the accuracy of predicted CO concentrations over remote rural areas, as well as the high CO concentrations over mountain areas around urban basins in the Shanxi province".

**Question**: (4) Probably, in the title, 'A comparative analysis for a deep learning model …' is more appropriate, as only one deep learning model is used in this study.

**Answer**: The title has been changed.

Reviewer #2:

In this manuscript, the authors evaluated the performance of a deep learning model and Kalman Filter in predicting CO concentration in China. The topic is interesting and fits the readership of GMD. However, the manuscript lacks details in the description of method, which hinders understanding of the results. A more thorough literature review of the performance of CO simulation and assimilation is also needed. Additional experiments should also be performed to demonstrate whether the DL model has good performance in predicting CO, see specific comments below.

**Answer**: Thank you for the comments! The manuscript has been revised based on the comments.

**Question**: Title: please clarify whether it is CO emission or concentration.

**Answer**: We are focusing on the prediction of CO concentrations. The title has been changed.

**Question**: L20-21: please clarify whether the statistics in E. China come from the training data. It is not clear to me what is the difference between E. China and the independent observation just based on the description here.

**Answer**: The abstract has been revised: "We find the performance of DL model is better than KF in the training period (2015-2018): the mean bias and correlation coefficients are 9.6 ppb and 0.98 over E. China, and are -12.5 ppb and 0.96 over grids with independent observations (i.e., grids with CO observations that are not used in DL training and KF assimilation)".

**Question**: L27: what is the cause of this? Does DL model have less capability in capturing the extreme values?

**Answer**: As shown in the revised manuscript: "As shown in Fig.4A-B, extreme pollution events, with CO concentrations > 1200 ppb, account only 3.4% of the total number of observations. It cannot be learned sufficiently, because the DL model, as a data-driven approach, would require more observations about the extreme pollution events to improve the predictions".

**Question**: L38-43: since this manuscript is focused on CO, I don't see it necessary to discuss NO2, PM2.5, and O3 here. Instead, readers are more interested in what the

performance of CO simulations are compared to different observations. L45-49: similar comment as the previous one. Data assimilation of other species is not relevant here. What are previous data assimilation efforts for CO? There are many of them and should be summarized here. L71: I think you should introduce the focused species right at the beginning. Also, why choose CO for this comparison? Can the conclusions for CO be extended to other species considering their similarities and differences?

**Answer**: Thank you for pointing out this issue! The Introduction Section and references in the Introduction Section have been revised. We believe the conclusions for CO can be extended. As shown in the revised manuscript: "We assume comparable or better performances of DL in the predictions of $O_3$ and $PM_{2.5}$ than the CO analysis shown in this work, because of their shorter lifetimes".

**Question**: L100: what is the total number of grids?

**Answer**: The assimilation domain is E. Asia (Fig. 2). The total number of grids in the assimilation domain is about 5000.

**Question**: L154: cost function is not defined.

**Answer**: The definition of the cost function was added in Section 2.3.

**Question**: L158-160: it is not clear what is being done here. Please clarify.

**Answer**: The description has been updated: "For a faster convergence speed and the stability of the model performance, we rescale all the features to a nearly same scale. The processing method is multiplying the original variable by a constant $10^n$ and adapting $n$ for each variable according to the specified scale. For example, most of the values of sea level pressure (SLP) are distributed around $10^5$, so we multiply SLP by $10^{-4}$ to make the value of the feature SLP distributed around $10^1$". We are sorry for the confusion.

**Question**: Section 3.1: It seems to me that DL can capture the temporal variations in 2019-2020 because the seasonality is unchanged compared to the training data, but when the magnitude changes in the test period (2019-2020), DL cannot capture. I am still doubtful how much information one can get from this DL model.

**Answer**: Thank you for pointing out this point! The manuscript has been revised to discuss the possible sources for the larger discrepancy in 2019-2020: "The rapid decrease of surface CO concentrations over NCP 2019 could be associated with an unexpected drop in CO emissions, which is not considered in the linear projection of emission inventory, and led to overestimated CO concentrations in the DL model. In addition, recent studies (Li, K. et al., 2020; Chen, X. et al., 2021) indicated a dramatic increase in surface O3 concentration over NCP in 2019. The possible changes in atmospheric oxidation capability and sink of CO may not be sufficiently captured by the DL model, as the relevant information is not used as the input while training the model".

**Question**: Fig2 & 3: please clarify are these the 90% training stations?

**Answer**: Fig. 2 and Fig. 3 are produced based on the 90% stations. The captions of these two figures have been revised to clarify this point.

**Question**: L213: I don't think the performance in 2015-2018 is even worth mentioning if you are using the 90% training stations, since this is the training data. A good consistency during this period does not mean anything regarding the model performance in prediction.

**Answer**: This sentence has been removed.

**Question**: L230-231: If that is the case, you should evaluate the model using only 2019 data to demonstrate how well the model performs in a normal case, and then another evaluation for only early 2020 to show how much emission reductions affect the prediction.

**Answer**: Thank you for this suggestion! The manuscript has been revised: "As shown in Fig. 3F, the MEE CO observations are about 10.2% and 25.8% lower than predicted CO by DL model in Feb 2019 and 2020, respectively; the MEE CO observations are about 11.1% and 14.2% lower than predicted CO by DL model in Jun-Aug 2019 and 2020, respectively. Assuming the difference in Jun-Aug (i.e., 11.1% and 14.2%) represents the annual CO emission trends, our analysis thus, suggests about 12.5% decline in CO emissions caused by COVID-19 controls, which is consistent with Gaubert et al. (2021)".

**Question**: L247: what is the sensitivity of KF results to model and observation errors? Is the poor performance of KF caused by the inappropriately assumed errors?

**Answer**: We have assumed a large model error (50%) and a small observation error (1%) to ensure the modeled CO concentrations can be adjusted sufficiently in KF. It led to a dramatic improvement in the modeled CO concentrations, for example, the bias in the CO concentrations over E. China is reduced from 410 ppb to 115 ppb, by about 72%. More adjustments to the model and observation errors may beyond the limit in actual data assimilation applications.

**Question**: L259: what is the difference between KF in the training and testing period? Did you also feed in no observations for KF during the testing period? If that is the case, I don't even think this should be called KF, since the simulation purely depends on GEOS-Chem and initial condition.

**Answer**: The actual CO observations are assimilated in both 2015-2018 and 2019-2020 in the KF. The manuscript has been revised to clarify this point: "In contrast to the DL approach, CO observations are assimilated in KF in both periods".